# Valproic Acid in Pregnancy Revisited: Neurobehavioral, Biochemical and Molecular Changes Affecting the Embryo and Fetus in Humans and in Animals: A Narrative Review

**DOI:** 10.3390/ijms25010390

**Published:** 2023-12-27

**Authors:** Asher Ornoy, Boniface Echefu, Maria Becker

**Affiliations:** 1Department of Morphological Sciences and Teratology, Adelson School of Medicine, Ariel University, Ariel 40700, Israel; bonifacee@ariel.ac.il (B.E.); mariabe@ariel.ac.il (M.B.); 2Department of Medical Neurobiology, Hebrew University Hadassah Medical School, Jerusalem 9112102, Israel

**Keywords:** valproic acid, teratogenic antiepileptic drug, neural tube defects, autism

## Abstract

Valproic acid (VPA) is a very effective anticonvulsant and mood stabilizer with relatively few side effects. Being an epigenetic modulator, it undergoes clinical trials for the treatment of advanced prostatic and breast cancer. However, in pregnancy, it seems to be the most teratogenic antiepileptic drug. Among the proven effects are congenital malformations in about 10%. The more common congenital malformations are neural tube defects, cardiac anomalies, urogenital malformations including hypospadias, skeletal malformations and orofacial clefts. These effects are dose related; daily doses below 600 mg have a limited teratogenic potential. VPA, when added to other anti-seizure medications, increases the malformations rate. It induces malformations even when taken for indications other than epilepsy, adding to the data that epilepsy is not responsible for the teratogenic effects. VPA increases the rate of neurodevelopmental problems causing reduced cognitive abilities and language impairment. It also increases the prevalence of specific neurodevelopmental syndromes like autism (ASD) and Attention Deficit Hyperactivity Disorder (ADHD). High doses of folic acid administered prior to and during pregnancy might alleviate some of the teratogenic effect of VPA and other AEDs. Several teratogenic mechanisms are proposed for VPA, but the most important mechanisms seem to be its effects on the metabolism of folate, SAMe and histones, thus affecting DNA methylation. VPA crosses the human placenta and was found at higher concentrations in fetal blood. Its concentrations in milk are low, therefore nursing is permitted. Animal studies generally recapitulate human data.

## 1. Introduction

Although the most sensitive period to teratogens is during active organogenesis at post conception weeks 3–8, there are several organs (i.e., teeth, external genitalia, brain) that continue to be very active developmentally beyond that period and may therefore still be affected by teratogens. Moreover, many drugs, especially antiepileptic drugs (AEDs) and drugs affecting mood are to be taken throughout pregnancy, thus possibly affecting the conceptus before and after the period of active organogenesis [1,2,3,4]. Therefore, teratogenic drugs that are taken for long periods of time are generally more hazardous compared to drugs administered for a limited time.

AEDs are used to control various types of convulsive disorders; some of them are administered as mood stabilizers or are sometimes administered for migraine and neuralgia treatment [3,4,5]. In addition to the anatomic and functional damage that can be caused by many of these drugs, AEDs also have the potential to induce in the offspring neurological, behavioral and cognitive effects.

In the majority of epileptic women planning a pregnancy, antiepileptic drugs cannot be discontinued because of the risk of seizures during pregnancy, which can be harmful to both mother and child [4,6]. Due to variations in the rate and degree of the teratogenic potential of the different AEDs, it is generally advised to use the less teratogenic drugs in the minimal effective dose during pregnancy.

There seems to be sufficient data to imply that valproic acid (VPA) is a highly teratogenic drug, apparently the most teratogenic antiepileptic drug [6,7,8,9,10,11]. There are also studies showing that the different types of epilepsy (if untreated) are not teratogenic [6,12]. This is further demonstrated by the use of some non-teratogenic antiepileptic drugs (i.e., lamotrigine) that do not increase the rate of congenital malformations, indicating that the teratogenic effects of the AEDs result from the direct effects of the drugs on the developing embryo and fetus. Moreover, VPA administered for the treatment of bipolar disorder also resulted in increased malformation rates [6,7], inducing a significant change in the guidelines for VPA prescription [13]. Many of these drugs may also cause withdrawal symptoms in the newborn infant.

The contrasting faces of VPA, many therapeutic benefits as opposed to the high percent of rather severe abnormalities if taken during pregnancy, remind us of the well—known Novella written by Robert Louis Stevenson and published in 1886, “**Strange Case of Dr. Jekyll and Mr. Hyde**”, in which Dr. Jekyll is the good and perfect man (a very effective drug) but may become Mr. Hyde (if taken during pregnancy), the ultimate evil and a murderer.

We searched in PUBMED and other sources for studies on the effects of VPA in pregnancy in humans and in animals. Many of the studies assessed the data related to several AEDs. For the evaluation of the rate of major congenital malformations (MCMs) or of developmental problems in the current review we used only studies that also described, among the antiepileptic drugs, the results of VPA exposure in pregnancy. As there is a very large number of studies related to the VPA-induced congenital malformations, especially in the last 20 years, we chose to summarize mainly prospective studies that compare the VPA data with controls or with other AEDs. A large proportion of these studies have been cited in our previous review on this subject [6].

The more teratogenic commonly used AEDs seem to be valproate, phenytoin, phenobarbital and carbamazepine [4,6,10].

Embryonic and fetal damage caused by AEDs may be manifested as follows:

A. An increased rate of major congenital malformations of many organs including heart, limbs and brain [6,14,15,16].

B. Specific syndromes, such as “Valproate syndrome” or “Anti-epileptic drug syndrome”, mainly affecting the cranio-facial complex, causing facial dysmorphism, and generally accompanied by neurological dysfunction and developmental delay [11,17,18,19]. Often, there are also malformations of other organs. The more accurate name for VPA-induced malformations is apparently “fetal valproate spectrum disorder” [20].

C. Neurodevelopmental disorders, mainly affecting cognitive function and behavior [21,22,23]. These changes may affect language development, learning abilities and even cause Autism Spectrum Disorder (ASD) or Attention Deficit Hyperactivity Disorder (ADHD) as well as learning difficulties [24,25,26].

Since many of the VPA-induced effects in human pregnancies were also demonstrated in experimental animals, especially rodents, we also summarized many of the animal data related to major malformations and neurobehavioral deviations. In addition, in order to understand the mechanism/s of VPA-induced teratogenicity we also summarized existing data on changes in gene expression and data on possible mechanisms of VPA teratogenic action. Finally, we also dealt with literature describing attempts to prevent the teratogenic effects of VPA, especially those that antagonize the epigenetic effects of VPA on the fetal nervous system.

## 2. The Therapeutic Effects of VPA

Valproic acid (2-propylpentanoic acid) is a short-chain fatty acid and is one of the substances in the group of histone deacetylase inhibitors (HDACi).

VPA has been on the market as an anticonvulsant since 1974, and is used in many countries because of its efficiency against several types of epilepsy and as a mood stabilizer. One of its main actions is the increase in the level of gamma amino butyric acid (GABA) in the brain. GABA, an inhibitory neurotransmitter, is an important inhibitor of seizures, as a reduction in GABA levels may potentiate seizures. For seizure control, the daily doses range between 300 mg and 2 gr, aiming to achieve therapeutic plasma levels of 50–100 microgram/mL. The lower doses are usually administered in the treatment of bipolar disorder, for manic patients, and in the treatment of migraines.

VPA has several additional and important beneficial effects. It is a potent epigenetic modulator and as such is used as an anticancer therapeutic agent in advanced prostatic cancer and in breast cancer [6,27,28]. Moreover, VPA also has reno-protective effects in diabetic nephropathy [28], protects the kidneys from acute renal ischemia reperfusion damage, has neuroprotective effects enhancing neuronal repair after stroke, and even has some antifungal and antimicrobial effects.

## 3. Possible Damage and Major Congenital Malformations (MCMs) Caused by VPA in Pregnancy (Table 1)

There are a number of ongoing international registries regarding the use of AEDs during pregnancy. Among the largest are The North American Antiepileptic Drug registry (AED Pregnancy Registry—https://www.aedpregnancyregistry.org), the North American Antiepileptic Drug Pregnancy Registry by the National Institutes of Health (.gov) (https://pubmed.ncbi.nlm.nih.gov), the EURAP—International Registry of Antiepileptic Drugs (https://eurapinternational.org) and the [MI] Medicines and Pregnancy Registry—Antiepileptic use NHS Digital in the UK (https://digital.nhs.uk › publications › statistical › antie) and the Australian Pregnancy Register For Women on Antiepileptic Medications (ABN: 38 128 668 797). Each of these registries has published their findings on the high teratogenicity of VPA and many of their studies are discussed in this review together with studies by other investigators (Table 1). To the best of our knowledge, all registries found high VPA teratogenicity with negative neurodevelopmental effects.

**Table 1 ijms-25-00390-t001:** VPA-induced malformations in humans.

Authors, Ref. Number	Main Findings	Comments
Jentink, et al. [3]	122 VPA-exposed children. VPA exposure resulted in spina bifida, atrial septal defect, hypospadias, polydactyly and craniosynostosis.	Use of VPA monotherapy was associated with significantly increased risks for six of the fourteen malformations considered.
Diav-Citrin et al. [7]	Multiple MCMs but no NTD. No increase in malformations in doses below 1000 mg/day. Also increased MCMs in treated non-epileptics.	Polytherapy lead to a fourfold increased rate of teratogenicity. The incidence of MCMs was lower in monotherapy.
Weston et al. [10]	Children exposed to VPA (467) were at a higher risk of malformation compared with children born to women without treatment. At doses <1500, the rate of MCMs was 10.4%.	Prospective, cohort and randomized control studies. Several studies with contrasting results on VPA outcome relative to dosage.
Kini, et al. [11]	Of a total of 63 children exposed to valproate in utero, 14% had MCMs.	Children exposed to VPA have distinctive facial features of VPA syndrome.
Kozma, [18]	In 69 cases of VPA-induced MCMs, 62% had musculoskeletal malformations, 26% had cardiovascular anomalies, 22% had genital anomalies, 16% had pulmonary anomalies and 30% had minor skin defects.	VPA monotherapy with typical phenotypic features.
Mutlu-Albayrak, et al. [19]	Four children with dysmorphism, bilateral cryptorchidism and skeletal malformations.	Study conducted on two pairs of siblings
Bromley, et al. [22]	In 36% of children with valproate syndrome, NTD and different malformations were observed.	Cochrane comprehensive review.
Bromley et al. [23])	Spinal, skeletal, cardiac and facial malformations.	The rate of malformation correlated with the dose of VPA.
Verloes, et al. [29]	Dysmorphism and multiple malformations including upper limbs, kidneys and brain.	A child with multiple malformations probably associated with VPA.
Bromfield, et al. [30]	284 VPA-exposed pregnancies resulted in 30 (11.0%) offspring associated with MCMs.	Dose response: an increased dose of VPA was associated with a higher incidence of malformation.
Tanoshima, et al. [31]	Cumulative meta-analyses of cohort studies on VPA-associated MCMs revealed NTD, genitourinary musculoskeletal, cleft lip and palate and congenital heart malformations.	VPA should not be used as a first-line therapy in women of childbearing age. VPA-induced MCMs are 2–7-fold higher than with other AEDs.
Eadie, M. J. and F. J. Vajda [32]	At very high VPA doses, above 1400 mg/day, fetal malformation increased to 33.9% of 172 exposed pregnancies.	The rate of MCMs is the same in all doses below 1400 mg VPA.
Campbell, et al. [33]	1290 women exposed to VPA resulting in MCMs in 6.7% in a dose-dependent manner. At doses lower than 600 mg there was no significant increase in MCMs.	VPA is more teratogenic compared to other AEDs.
Artama, et al. [34]	The 65 MCMs recorded in 1411 patients were higher in the offspring of epileptic patients using VPA than untreated.	Highest risk following VPA exposure compared to carbamazepine, oxcarbazepine, or phenytoin.
DiLiberti et al. [17]	Malformations known to be associated with valproate syndrome, including autistic features.	Early study of seven cases of VPA exposure.
Yonkers et al. [35]	Expert panel reviewed articles on the management of bipolar disorder using AEDs and discovered a 5% to 9% occurrence of VPA-mediated lumbosacral NTD.	The authors drafted a consensus document and outlined the consequences of the use of mood stabilizers in bipolar patients during pregnancy.
Hernandez-Diaz et al. [36]	A prospective study of 323 VPA exposed children between 1997 and 2013, finding MCMs in 9.3% (30 of 323).	VPA and phenobarbital were associated with a higher risk of MCMs than newer AEDs such as lamotrigine and levetiracetam.
Vajda et al. [37]	Dose-related increased incidences of MCMs: 43 malformed of 290 VPA-exposed infants.	No evidence that pre-conception folate supplementation reduced the hazard of AEDs-associated fetal malformation.

In humans, several Major Congenital Malformations (MCMs) are attributed to VPA teratogenicity and have been encoded under the phrase, “Valproate syndrome” that was mainly defined due to typical craniofacial dysmorphism [6]. There were arguments as to whether these malformations are VPA-induced or due to confounding factors, as pregnant patients tend to use multiple AEDs, and secondly, because these types of MCMs were similar to those previously reported in carbamazepine and phenytoin monotherapies [2,4,6,38,39]. Neural Tube Defects (NTD) were observed in 1–2% of VPA-exposed infants [23,40,41]. Cardiovascular, craniofacial and orofacial malformations, skeletal malformations and limb reduction abnormalities, rib and phalangeal malformations, tracheomalacia, urogenital malformations, hypospadias and visual and hearing deficits were all described [7,10,23,29,42,43]. In addition, a high rate of spontaneous abortions [44] and reduced neonatal weight [7] were reported. Enlarged cerebral ventricles, hypoplasia of the corpus callosum and an abnormal septum pellucidum are only part of the brain malformations. This may often cause severe neurodevelopmental problems, but neurodevelopmental abnormalities are often observed without any distinct malformations in the brain or other organs (i.e., ASD, ADHD intellectual impairment ext.) [8,40,44,45,46,47,48,49].

Many, but not all, of these malformations have been found in experimental animal studies as well. In rodents, as detailed later, it is mainly associated with exencephaly alongside malformations of various organs.

The first reports suggesting the teratogenicity of valproic acid in humans, increasing the rate of lumbosacral meningomyelocele or meningocele in children whose mothers were treated with VPA in the first trimester of pregnancy were published in the early 1980s and later by many other investigators [41,50,51,52]. In a French case–control study, the calculated odds ratio for the association of spina bifida after exposure to valproate was 20.6 [41,53]. Lindhout et al. found [52] that most malformed infants with NTD had open defects of the spinal cord, in addition to a high rate of other midline defects.

In the Swedish Medical Birth Registry [48], MCMs were reported in children (9.7%) born to women treated with valproic acid monotherapy early in pregnancy. The rate of MCMs was significantly higher than that found among infants whose mothers were treated with other anticonvulsants. A rate of 10.7% was also described in the North American Antiepileptic Drug Pregnancy Registry [48]. Bromfield et al. [30] found an MCM rate of 11% among 284 VPA exposed pregnancies, with the rate being similar in those with genetically determined epilepsy (idiopathic epilepsy) and those with epilepsy that is non-genetic in origin. Additional studies reported a similar rate of MCMs among offspring of children prenatally exposed to VPA [18].

Tanoshima et al. [31] analyzed 59 studies on the effects of VPA in pregnancy and found a 2–7—fold increase in MCMs over the rate observed from other AEDs. The main increase was in congenital cardiac anomalies, orofacial, clefts, genitourinary and musculoskeletal malformations. In a meta-analysis based on 1740 exposed infants reported in 11 controlled cohort studies with first trimester exposure to VPA, a 2.59-fold increase in the rate of major malformations was found when compared to all other antiepileptic drugs and a 3.77-fold increase was found when compared to the general population [45].

In a study performed by us [7] on the outcome of 154 VPA-treated pregnancies compared to 1315 control pregnant women, the rate of major anomalies, after exclusion of genetic disorders, was 6.7% as opposed to 2.5% in controls (RR of 2.66, 95% CI 1.25, 5.65). Three of the offspring with anomalies were suspected as having “valproate syndrome”. VPA polytherapy increased the teratogenic risk. VPA doses of less than 1000 mg were not associated with an increased risk of major anomalies. Malm et al. [47] suggested an additional hereditary effect on the rate and type of VPA-induced MCMs explaining the differences between studies.

## 4. VPA and Dose-Related MCMs

Eadie and Vajda [32] reported in a study from the Australian antiepileptic drugs registry that in 172 pregnancies exposed in utero to VPA monotherapy, the rate of MCMs was as high as 15.1%. The rate of malformations increased steeply with daily doses above 1400 mg. Campbell et al. [33] assessed the dose response of prenatal VPA teratogenicity and found that in daily doses of less than 600 mg, the rate of malformations was 5.0%, in daily doses of 600–1000 mg the rate increased to 6.1% but in doses above 1000 mg it was 10.4%. Such a dose response of VPA teratogenicity was also reported by other investigators with an increasing rate of congenital abnormalities in doses above 1000 mg/day [7,8,34,54].

## 5. VPA and Polytherapy

VPA Polytherapy also increases the rate of MCMs compared to monotherapy. When an antiepileptic medication is given in combination with other AEDs it may be difficult to point exactly to the drug with the major contribution to the abnormal outcome. This may also relate to the combinations of VPA with other AEDs, raising the question as to which is the major contributor. However, there are a number of studies demonstrating that the contribution of VPA to abnormal outcomes in pregnancies treated with several AEDs is a major one [55,56,57].

Holmes et al. [58] found that with the addition of VPA to lamotrigine, the rate of congenital malformations increased to 9.1% as compared to lamotrigine alone, which was only 2.9%. However, the rate was not increased when other AEDs, except VPA, were added to lamotrigine treatment. Similarly, the risk of malformations in children exposed to VPA and carbamazepine was 15.4%, while carbamazepine with any other AED was only 2.5%. Thus, VPA seems to be the main drug that has synergistic effects with other AEDs.

In a study that pooled data from five European prospective studies, Samren et al. [59] found that the rate of MCMs among the offspring of epileptic mothers treated only with VPA was 9%, and if VPA was given with other AEDs it raised the rate to 15.3%. GlaxoSmithKline (GSK) [56] reported, in a prospective follow-up of about 1100 pregnancies treated with lamotrigine in the first trimester of pregnancy, a rate of major congenital anomalies of 2.4% (27/1124), similar to controls. Exposure to lamotrigine with another AED excluding VPA did not change significantly the rate of anomalies. However, the exposure to lamotrigine and VPA raised the rate of major anomalies to 11.0% (14/127) pointing to the major role of VPA in this increase. Morrow et al. [8], in the UK study on the effects of AEDs polytherapy in pregnancy, found that the highest rate of anomalies was among the offspring of mothers treated with VPA together with other AEDs.

In a meta-analysis we performed on the effects of carbamazepine (CBZ) treatment in pregnancy on 1155 pregnancies [55], the addition of VPA to the carbamazepine increased the rate of MCMs from 5.5% following carbamazepine exposure to 11%, while in the non-exposed controls it was only 2.3%. Tomson et al. [60] observed, in the prospective EURAP study of AEDs, an increase from 10.0% of malformations following VPA monotherapy (observed in 1224 pregnancies) to 11.3% with the addition of lamotrigine to VPA treatment (159 pregnancies). The addition of AEDs other than lamotrigine (205 pregnancies) raised the rate of malformations to 11.7%. A possible explanation for this observation may be in the findings reported by Zaccara and Perrucca [61], which showed a rise in the blood levels of lamotrigine and phenobarbital induced by concomitant administration of VPA.

## 6. Valproic Acid Syndrome (Valproic Acid Spectrum Disorder)

A specific set of facial dysmorphic features related to the effects of VPA on the developing embryo and fetus was first described by DiLiberti et al. in 1984, in seven infants [17]. This syndrome was later corroborated by many authors describing additional children exposed in utero to VPA and exhibiting similar facial features [11,18,21,62,63,64]. The main clinical findings include intrauterine growth retardation (IUGR), a long and thin upper lip, shallow philtrum, epicanthal folds and mid face hypoplasia manifested by a flat nasal bridge, small upturned nose and downturned angles of the mouth (Figure 1). Many of the children with “valproate syndrome” also have other congenital anomalies, developmental delay and neurological impairment [21]. A similar set of anomalies was also described in the offspring of women using other AEDs in pregnancy, with definitions representing quite similar facial dysmorphic features following in utero exposure to phenytoin, carbamazepine, phenobarbital, trimethadione and primidone [40,63]. Thus, it is accepted that these dysmorphic features constitute the “Antiepileptic Drugs Syndrome”.

Dean et al. [65] proposed a set of clinical findings that might be of help in diagnosing the “fetal antiepileptic drug syndrome” and they also included facial dysmorphic features that seem to be common to all antiepileptic drugs. Kini et al. [11] found that among fifty-six children exposed in utero to VPA, only five (9%) had normal facial appearance while twenty-four (42%) had moderate to severe facial dysmorphic features similar to those described for the VPA syndrome. A reduction in verbal intelligence was common among these children. Based on many studies, it seems that the “fetal valproate syndrome” is not significantly distinct from the “antiepileptic drug syndrome. It is difficult to diagnose VPA embryopathy from the facial appearance only, as similar facial features can be observed with other AEDs and also in a small percentage of normal non-exposed children, without a history of intrauterine VPA exposure. The specific facial features, often accompanied by other major anomalies and/or developmental delay, are a helpful tool in the diagnosis of VPA embryopathy [11,40,63,65]. Indeed, taking all of the data together, facial dysmorphic features, major congenital malformations and neurodevelopmental deficits, Clayton Smith et al. [20] coined the term “Fetal Valproate Spectrum Disorder” defining all of the characteristics of the possible damage induced by prenatal VPA exposure in their consensus European statement.

The risk for major congenital malformations in the offspring of VPA-treated mothers is about three times higher than in the non-exposed population of children, being about 10% (Table 1). Doses below 600 mg/day seem to have low teratogenicity while daily doses above 1000 mg/day are highly teratogenic; in the highest doses the rate even exceeds 10%. Many of the VPA-exposed children also present the typical “valproic acid spectrum disorder” and may have neurodevelopmental problems. If therapy with VPA is mandatory, the lowest effective daily dose should be administered, divided into three doses to minimize fluctuations in serum levels, thus reducing the risk to the fetus. VPA polytherapy, especially the combination of VPA with lamotrigine or carbamazepine, seems to increase the teratogenic effects of these drugs.

## 7. VPA and Neurodevelopmental Problems

### 7.1. Impaired Neurodevelopment

Impaired neurodevelopment induced by prenatal VPA is based on prospective and retrospective neurodevelopmental studies of a relatively large number of children prenatally exposed to VPA. Neurodevelopmental deficits, including delayed motor skills, reduced cognitive functions, language impairment, specific neurodevelopmental syndromes (i.e., ASD and ADHD) and behavioral deficits were all described [6,9,24,66,67]. There seems to be a strong correlation between the dysmorphic facial features constituting the valproate syndrome and neurodevelopmental/cognitive impairment.

While evaluating the results of developmental studies, it is important to remember that there may be many confounding factors that should also be considered. Some of them, like alcohol or drug abuse, are not reported. Hence, the presence of the typical facial features of valproate syndrome is an important marker for the involvement of VPA in neurodevelopmental problems.

### 7.2. Intellectual and Learning Disabilities 

As of today, the intellectual abilities of hundreds of children exposed in utero to VPA have been studied [11,24,46,63,65,66,67,68,69,70,71,72,73,74,75]. Koch et al. [66] studied 40 children exposed in utero to a single antiepileptic drug: phenobarbital, phenytoin or VPA. The VPA-exposed neonates exhibited high excitability and the degree of excitation correlated with the VPA serum concentrations at birth. Later, at six years of age, the neurological dysfunction of these children further correlated with their neonatal VPA blood levels [66]. Kini et al. [11] studied 63 children, 0.5–16 years of age who were exposed in utero to VPA, and found that 14% had major anomalies. Most of the exposed children had distinct facial features of valproate syndrome as well as decreased verbal intelligence. Moore et al. [68] found that, among 34 children who were exposed in utero to VPA monotherapy, many had speech developmental delay and behavioral changes, with four having autistic features. Dean et al. [65] found that of 47 children exposed in utero to VPA, 40% had distinct facial features of VPA embryopathy, and 28% (13) had developmental delay. Viinikainen et al. [70] found that of 13 children prenatally exposed to VPA monotherapy, 8 (62%) required educational support. Of these, four children had major neurobehavioral problems but none of the thirteen controls did. The other nine had minor problems compared to five with minor problems among the controls. Gaily et al. [74] found that while carbamazepine exposure in utero was not associated with developmental delay, the exposure to carbamazepine and VPA was associated with significant developmental delay, especially in verbal intelligence. Adab et al. [71,72] found that among 41 children exposed to VPA monotherapy, there was a significant reduction in mean verbal IQ. This was not observed in carbamazepine- or phenytoin-treated children.

One of the largest studies was carried out by Daugaard et al. [76] (84) on a population of 913,302 children. Of these, 580 children were prenatally exposed to VPA. There was a significant increase in intellectual disabilities in the VPA-exposed children, with an adjusted odds ratio of 4.48 (CI 2.97–6.76) compared to non-exposed children (Table 2).

Shallcross et al. [78], in the UK, studied the neurodevelopment of 44 VPA-exposed infants of less than 24 months of age in comparison to 51 infants born to mothers treated with levetiracetam and 97 controls using the Griffith Mental Development Scales. They found normal development in infants prenatally exposed to levetiracetam, but reduced mental abilities in those born to VPA-treated mothers. [78]. Similarly, Bromley et al. [9] summarized several prospective studies and found that the IQ of 123 young children exposed prenatally to VPA with or without the typical facial dysmorphic features was 8.72 points lower than in children born to untreated epileptic women. In a subsequent study Bromley et al. [22] reported that among 31 children prenatally exposed to VPA exhibiting the typical facial features of VPA embryopathy, the intellectual function was lower by 19 IQ points compared to controls, with 26% having an IQ lower than 70. Verbal comprehension was lower compared to the other cognitive functions. These data were similar in children that had or did not have additional MCMs (Table 2).

Cohen et al. [25] found, in a comparative study on the learning and memory abilities of 6-year-old children prenatally exposed to either lamotrigine, carbamazepine, phenytoin or VPA [25], that children prenatally exposed to VPA exhibited significant difficulties in their abilities to “process, encode and learn auditory/verbal, and visual/non-verbal data”.

It is obvious that prenatal exposure to VPA is associated in many children with a significant reduction in intellectual abilities, more prominent in verbal abilities, and with increased learning difficulties. This reduction is apparently not related to the existence of other MCMs, but is significantly higher if there are specific facial features of VPA embryopathy.

### 7.3. VPA and ADHD

During the last years, information has accumulated that prenatal VPA not only increases the rate of ASD among the offspring, but also significantly increases the prevalence of ADHD. In our study [24] on 30 children exposed prenatally to VPA compared to 42 children exposed to lamotrigine and 52 non-exposed children we used the Conners questionnaire [79] to assess preschool ADHD. Higher scores imply a higher probability of ADHD. The global score of the parents Conners questionnaire was significantly higher (*p* = 0.01) than that of lamotrigine-exposed children or of controls. The ADHD score of the teacher’s questionnaire was also higher, but the difference was insignificant. Similarly, visual–motor integration, visual perception and motor control were significantly lower. None of the VPA-exposed children had facial dysmorphic features.

Cohen et al., 2013 studied the rate of inattentive and combined types of ADHD in 45, six-year-old children prenatally exposed to VPA in comparison to children exposed to other AEDs [25]. They found that 10 of the children (22%) had either inattentive or combined types of ADHD according to the parents’ BASC questionnaire and 11 (26%) had either inattentive or combined types of ADHD according to the teachers’ BASC questionnaire.

Results of studies on much larger cohorts of VPA-exposed children further verified these findings. Christensen et al. studied the occurrence of ADHD among 580 children prenatally exposed to VPA in comparison to 912,772 unexposed children and found a rate of 8.4% of ADHD among the exposed children and only 3.2% in the unexposed [26]. Other antiepileptic drugs did not increase the rate of ADHD among the prenatally exposed children. Wiggs et al. found, in a cohort of 701 children of epileptic mothers prenatally exposed to VPA, a hazard ratio for ADHD of 1.74 according to DSM 10 criteria [80]. No significant association was found following prenatal exposure to carbamazepine or lamotrigine.

Relatively few studies, mainly in recent years, evaluated the associations between prenatal exposure to VPA and ADHD. However, it seems clear from those studies that VPA in pregnancy increased the risk of ADHD in the offspring.

### 7.4. VPA and Autism Spectrum Disorder (ASD)

A possible association between in utero VPA exposure and ASD was apparently first observed by Christianson et al. who described four children exposed in utero to VPA; all demonstrated developmental delay and one of these children also had ASD [68]. Later, Williams and Hesh [81] and Rasalam et al. [82] described additional children with the typical facial features of VPA embryopathy who also developed the typical findings of autism as outlined in the DSM IV. Moore et al. [63] found that among 57 children affected by antiepileptic drugs, four had ASD and two had Asperger syndrome. Five of the affected children (10.8% of 46 exposed to VPA) were exposed to VPA alone or combined with carbamazepine or phenytoin. Bromley et al. [83] investigated the impact of prenatal exposure to VPA and of other antiepileptic drugs on the rate of neurodevelopmental disorders in a cohort of children up to six years of age (n = 415). Prenatal exposure to VPA monotherapy increased the risk of ASD to 12% (6/50) and up to 15% (3/20) when used with other AEDs, compared with control children (4/214; 1.87%) and with children exposed to carbamazepine (1/50) or lamotrigine (2/30) [83]. It seems that the rate of ASD, according to the DSM IV criteria, including pervasive developmental disorder (PDD) and Asperger syndrome, among those exposed to VPA mono- or polytherapy was about 10%, about 20 times higher than the rate of ASD at that time (Table 3).

Other investigators, generally using the ICD 9 criteria, found a slightly lower rate of ASD, but one still significantly higher than in the general population. For example, Christensen et al. conducted a population-based study evaluating 655,615 children born from 1996 to 2006 in Denmark, with 508 prenatally exposed to VPA [84]. They found a clear association between prenatal exposure to VPA and an increased risk of childhood autism. However, the prevalence of ASD (ASD and childhood autism) was only about 4.42% with a hazard ratio of 2.9. Later, several studies reported similar associations of prenatal VPA treatment and ASD [80,85,86,87,88]. Wood et al. [85] determined a direct association with ASD and higher doses of prenatal VPA exposure in 6–8-year-old children who were recruited via the Australian Pregnancy Register for the Women on Antiepileptic Medication prospective study. Wiggs, et al., examined the associations of prenatal AEDs with the risk for ASD and ADHD in 14,614 children born between 1996 and 2011 to women with epilepsy exposed to the three most commonly used drugs (VPA, lamotrigine and carbamazepine), using the Swedish register data [80]. They found that the highest risk of ASD and ADHD was related to the maternal use of VPA. A slightly higher but insignificant risk compared to controls was found for carbamazepine, whereas exposure to lamotrigine had no evidence of risk. Honybun et al. [86] examined in 121 children in Australia, the effects of prenatal AED exposure in association with ASD and ADHD symptoms. While there was a preponderance of males exhibiting ASD symptoms over females in the entire group, the sex ratio differences were not observed among the 54 children exposed to VPA (Table 3).

**Table 3 ijms-25-00390-t003:** VPA and autism spectrum disorder.

Authors, Ref. Number	Main Findings	Comments
Moore et al. [63]	Among fifty-seven children affected by AEDs, four had ASD and two had Asperger syndrome; five of the affected children (10.8% of 46 exposed to VPA) were exposed to VPA alone or combined with carbamazepine or phenytoin.	VPA in utero is a key factor in ASD development.
Christianson et al. [68]	Four children that were exposed to VPA in utero; all demonstrated developmental delay and one of these children also had ASD.	First evidence of VPA association with developmental delay and ASD.
Rasalam et al. [82]	Five of fifty-six (8.9%) of the children exposed to VPA alone had autistic disorder.	Prenatal exposure to VPA is a risk factor for the development of ASD.
Williams and Hesh [81]	Case study of a male child who has a clinical phenotype with both fetal valproate syndrome and autism.	Suggested a relation between prenatal VPA and autism.
Wood et al. [85]	Prospective study: high doses of VPA exposure in utero were mostly associated with autistic traits.	Statistical evidence for VPA exposure in utero and ASD.
Wiggs, K.K.;et al. [80]	Prenatal VPA was associated with ASD (hazard ratio [HR] 2.30). No statistically significant association with ASD was reported for use of carbamazepine, and lamotrigine.	In utero VPA exposure is a high-risk factor for ASD development.
Honybun et al. [86]	No sex ratio differences in preponderance in ASD symptoms were observed among the 54 children exposed to VPA.	VPA exposure in utero was associated with ASD symptoms in a sex-independent manner.
Bromley et al. [83]	VPA monotherapy increased the risk of ASD to 12% (6/50) and up to 15% (3/20) when used with other AEDs, compared with control children (4/214; 1.87%) and with children exposed to carbamazepine (1/50) or lamotrigine (2/30).	ASD was diagnosed according to the DSM IV criteria; in those exposed to VPA mono- or polytherapy, ASD was diagnosed in about 10%, about 20 times higher than in the control population.
Petersen et al. [87]	In retrospective cohort studies, the risk of MCMs, neurodevelopmental and behavioral outcomes in the VPA-exposed was twice the risk of those exposed to other AEDs.	The risks of adverse maternal and child outcomes in women who continued antipsychotic use in pregnancy were not greater than in those who discontinued treatment before pregnancy.
Christensen et al. [84]	Population-based study on 508 prenatally exposed to VPA followed up to 14 years of age. A total of 4.42% had ASD with a 2.5% risk for childhood autism.	The hazard ratio compared to those not exposed was 2.9.

Prenatal exposure to VPA may result in a significant increase in the rate of ASD. The rate of ASD among the offspring ranges from 3% to 4% to over 10%. Even considering the fact that the prevalence of ASD rose in the last 20 years to about 1%, the observed elevated rate due to VPA is indeed alarming. It is interesting to note that the increased ratio of males over females, generally found in children with ASD, was not observed among the VPA-affected children.

## 8. VPA and Folic Acid Administration in Human

VPA and many antiepileptic drugs (phenytoin, barbiturates, carbamazepine and lamotrigine) may interfere with folic acid absorption or metabolism, possibly an additional cause for their induction of congenital anomalies [6,44]. It is therefore recommended to treat women on AEDs at preconception and in the first 2–3 months of pregnancy with folic acid, which protects humans from NTD and possibly cardiac and oro-facial malformations. Although the use of folic acid supplementation has been shown to generally decrease the incidence of NTD in humans, there is disagreement as to the benefit of folic acid in reducing the rate of AED-induced congenital malformations and NTD, especially following VPA intake [38,41,42,43,44,45]. Despite the uncertainty of effectiveness, it is recommended for women on AED therapy to take 4–5 mg/day of folic acid prior to any planned pregnancy. Pittschieler et al. [89] found that periconceptional folic acid significantly reduced the rate of spontaneous abortions and premature delivery in women treated with VPA and carbamazepine. They did not study the rate of congenital anomalies. Other studies were inconclusive as to the beneficial effects of folic acid even in reducing the rate of spontaneous abortions [90,91]. Vajda et al. [37], in a study on 2104 women treated during pregnancy with VPA, did not find any beneficial effects of 5 mg/day of folic acid prior to and during pregnancy on the prevention of VPA-related birth malformations.

## 9. VPA Transplacental Passage and Secretion into Milk and Semen

A basic principle for the action of any agent on the embryo and fetus is its ability to cross the placenta. VPA is known to cross the human placenta, and the clearance of VPA is increased during pregnancy [92,93]. Valproic acid levels in cord serum are often higher than in the mother and may be up to five times higher than the levels in maternal serum at term, with mean ratios of 1.4–2.4 and a very large variation [94]. These increased concentrations in the fetus have been attributed to a better binding in the fetal compartment than in the maternal. The high, possibly toxic concentrations in the fetus, may partially explain the high teratogenicity of this drug.

VPA is excreted into human milk in relatively low concentrations [95,96,97]. Available reports suggest that a suckling infant may ingest less than 5% of the weight-adjusted maternal daily dose [93]. Kacirova et al. [95], in a study carried out in 2021 women, examined the VPA concentrations in breast milk and newborn serum and found that the infant serum concentrations and maternal concentrations in milk are low, ranging from 0.01 to 1.61, with an average of 0.51. Hence, lactation by VPA-treated mothers is permissible.

VPA is excreted in the semen of rabbits and humans [98,99] and in high doses it may cause a reduction in sperm count that returns to normal with cessation of treatment or significant dose reduction. The clinical importance of these findings is apparently negligible.

## 10. Malformations in Children of Untreated Epileptic Women

Since many of the antiepileptic drugs are teratogenic in man and in experimental animals, the question often asked is whether epilepsy may cause an increase in the rate of MCMs without any relation to treatment. This is in spite of the fact that there are several AEDs that are not teratogenic and do not increase the rate of MCMs in prenatally exposed children (i.e., Lamotrigine, Levetiracetam). Moreover, the studies carried out in experimental animals, which demonstrate AED teratogenicity, are performed on non-epileptic animals. That epilepsy might play an important role in the teratogenicity of AEDs was shown by Shapiro et al. in 1976 [16], who analyzed the data from two cohorts of children born to treated and untreated epileptic mothers—the Finnish and American data that did not include women treated by VPA. The authors raised the possibility that the increased rate of malformations found in the offspring of epileptic mothers treated by phenytoin stems from maternal epilepsy since a similar rate of malformations (apparently major and minor malformations) was similar in children born to non-treated epileptic women. If the father had epilepsy, the rate of anomalies was intermediate between that of the epileptic and control mothers.

Kaaja et al. [100] prospectively studied the outcome of pregnancies in 988 epileptic women, of which 239 were not exposed during the first trimester of pregnancy to AEDs while the others were treated with various AEDs. They found at birth a 3.8% rate of MCMs among the offspring of the treated women and a rate of only 0.8% among the offspring of untreated epileptic women, with no correlation between the number of seizures during pregnancy and the outcome. Vajda et al. [37], analyzing the Australian prospective antiepileptic registry, found that in 83 women with untreated epilepsy, the rate of major congenital anomalies in their offspring was only 3.6% as opposed to 13.3% in 166 pregnancies exposed to VPA. Janz in his review [101] found that the percentage of malformations in untreated epileptic mothers was close to that of the control group (4.96% in 1311 historical (retrospective) cases studies and 4.97% in 181 prospectively studied cases), while the rate of MCMs in the offspring of treated epileptic women was twice as high.

In addition, several studies have found that when AEDs are administered for psychiatric indications (i.e., VPA or carbamazepine for the treatment of bipolar disorder) they are still teratogenic, with the rate of malformations being similar to that in children of treated epileptic mothers [7]. In addition, there are distinct differences in the rate of specific anomalies in the offspring of epileptic mothers in relation to the AEDs they use (i.e., increased rate of NTD only following CBZ and VPA treatment). Moreover, the studies carried out in experimental animals which demonstrate AED teratogenicity, are performed on non-epileptic animals.

While analyzing the results of the different studies we should remember that the untreated epileptic women are often those with less severe disease. It is still possible that in more severe epilepsy, if untreated, the rate of MCMs may be higher.

From the data on children born to untreated epileptic women and those born to epileptic women treated with non-teratogenic AEDs, it seems obvious that the increased rate of MCMs and/or neurodevelopmental disorders among the offspring of epileptic women treated with AEDs directly stems from the treatment. An additional proof is the fact that in children of non-epileptic women (i.e., bipolar disorder) treated by teratogenic AEDs, the rate of malformations is similar to that in treated epileptic mothers. The clear dose response regarding the AEDs’ teratogenic effects is additional proof for the teratogenicity of the various AEDs.

## 11. Animal Studies on VPA-Induced Teratogenicity 

### 11.1. VPA-Induced MCMs–General

Since the confirmation of VPA as a potent human teratogen, many studies employing different laboratory animal species (mice, rats, zebra fish, hamsters, drosophila, chicken, calf, rabbits and monkeys) of various strains have made extensive efforts to recapitulate the adverse effects of VPA reported in humans, in order to understand the mechanisms by which this drug mediates teratogenicity [102,103,104,105,106,107,108,109,110,111,112].

Similar to its effects in humans, VPA has exhibited dose-related and developmental-age-dependent teratogenic effects in various forms including, but not limited to, increased mortality rate, intrauterine growth retardation, gross craniofacial and skeletal anomalies, epigenetic aberrations and neurodevelopmental disorders, including ASD-like behaviors, in all animal species studied [113,114,115,116,117,118].

In most studies, VPA was teratogenic, but the teratogenic dose differed from that in humans. Most malformations, when categorized at the organ-system levels, largely showed features and confirmations of ASD, neural tube defects (NTD), cardiovascular malformations, urogenital anomalies, limb malformations, brain cell abnormalities and facial and reproductive organ malformations [103,106,117,119,120,121,122,123,124,125,126,127] (Table 4).

### 11.2. VPA-Induced General (Microscopic) Effects on the Nervous Tissue

VPA teratogenicity is manifested in brain tissue in different forms and magnitude, in a dose- and sex-related fashion. Mowery et al. [124] reported related abnormalities in the cerebellar nuclei of rodents prenatally exposed to VPA, with interpositus, fastigial and dentate nuclei of the cerebellum mostly affected. Hara et al., demonstrated in female mice offspring that exposure to VPA at GD 12.5 significantly decreased the number of Nissl-positive cells in the prefrontal cortex, supporting the dimorphism of VPA-induced social and communication deficits, and morphological changes in the somatosensory and prefrontal cortex leading to memory deficits [128]. In the parietal and occipital lobes of mouse cerebral cortex, VPA induced a characteristic asymmetry of parvalbumin (PV) cell reduction across hemispheres [129]. Ferret pups injected intraperitoneally with VPA, 200 μg/g on PND 6 and 7 (relatively late stage of corticogenesis), show that VPA causes an alteration in gyri formation specifically in the ferret’s frontal and parietotemporal cortical divisions [130]. VPA was administered to pregnant ICR mice as a single dose of 400 mg/kg on gestational days 6, 7, 8, or 9. The day 10 embryos revealed dose -dependent spinal nerve defects via a pathway that disrupts neural crest and somite formation but not related to the neural tube [102]. Damage to the cerebellar Purkinje cell layer was observed in young mice whose mothers were exposed to 400 mg/kg of VPA during pregnancy [121,128,131].

### 11.3. VPA-Induced Microscopic Pathological Changes in the Liver and Heart

VPA may induce hepatotoxicity and increased liver oxidative stress [132]. Hence, it is not surprising that it may causes hepatic damage/malformations in the VPA-exposed fetuses. One of the very few studies on the teratogenic effect of VPA on the liver reported that a single intraperitoneal injection of VPA (200 mg/kg) on GD 8 in pregnant mice resulted in offspring with a smaller liver size, whereby microscopic studies revealed a dilation of the central vein of liver lobules, breakage of the endothelial lining of the liver’s central veins, edematous swelling and a distortion of the normal architecture of the liver parenchyma [133]. Electron microscopic studies of the kidneys of mice prenatally exposed to VPA showed irregular glomerular basal membranes and marked deletions of the foot process [134,135]. Prenatal VPA induced the upregulation of thrombospondin-1 and activin proteins, inhibiting angiogenesis and vasculogenesis (manifested as failed endothelial cell tube formation) in a dose-related manner [123].

A significant increase in the rate of cardiac malformations or tissue damage at the microscopic level has been reported in fetal mice following prenatal exposure to VPA [136], with additional findings showing alterations in cardiac contractile function arising from an abnormal increase or a decrease in cardiomyocyte differentiation, giving rise to myocardial disorganization. This was dependent on the gestational age at the time of VPA exposure [39].

### 11.4. VPA-Induced Congenital Malformations in Animals

VPA administration during pregnancy alone or in combination with other drugs, induces a high number of anomalies, resorption and increased mortality rate. Most of the studies reported on exencephaly and spina bifida occulta as well as spina bifida aperta, thereby mimicking NTD as found in humans [6,48,103,114,115,116,125,137,138,139,140,141]. As expected, in addition to the dose relationship of VPA in the pathophysiology of NTD, gestational age is a crucial factor in determining what type of NTD is induced by VPA exposure. For example, Nau and co-workers treated mice with multiple high doses of VPA on gestation day 9 (GD) but not GD 8, and successfully produced posterior neural tube failure coinciding with spina bifida, unlike preceding studies where VPA on day 8 only produced cranial NTD (exencephaly) [125]. The rate of NTD was significantly reduced by high doses of folic or folinic acid [127]. Methionine, if given to pregnant mice a short time prior to VPA injection, significantly reduced embryonic VPA damage [142]. Similarly, vitamin E also decreased the rate of VPA-induced anomalies and embryonic damage in Balb/C mice [143], pointing to the possibility that oxidative stress is involved in VPA-induced embryonic damage. In NMRI mice, Vitamins B6 with B12, and to a lesser degree folinic acid, only partially reduced the rate of NTD, cleft palate and sternal malformations. Protection was incomplete, pointing to the possibility that VPA teratogenicity is not mainly through interference with folic acid metabolism [127,142]. High doses of VPA also produced intrauterine growth retardation and craniofacial and skeletal anomalies in rhesus monkeys [115]. It is important to note, however, that the doses of VPA used in animal models of NTDs are very high, equaling about 10 times the recommended human dose, and by implication point to the fact that human embryos are more prone to VPA toxicity than animals [6]. Prenatal VPA induces a variety of congenital malformations and fetal tissue damage in animals, similar to its effects in human. However, the tested animals seem to be less susceptible to the teratogenic effects of VPA, as higher doses are needed to elicit similar teratogenic effects to those in humans.

## 12. VPA-Induced Autistic-like Behavior in Animals 

Rodents exposed to VPA during prenatal development exhibit behavioral deficits that resemble the characteristics observed in individuals with autism [90,144,145,146], a fact that strengthens the possible association in man between in utero VPA exposure and ASD. Prenatal VPA exposure in animals has been proposed as a model of autism that has both construct and face validity due to similarity in disease etiology and resemblance to human symptoms [144,147].

Numerous studies and reviews highlight the impaired behavioral phenotype in rodents prenatally exposed to VPA. Decreased exploration associated with locomotor hyperactivity and stereotypic-like behaviors and reduced social interactions were mainly observed in VPA-induced ASD-like behavior in rodents [144,147,148,149,150,151,152,153] in a sex-dependent manner, which is often observed in individuals with autism [154] (Table 5).

Rodier et al. [155] found that an injection of 350 mg/kg body weight to rats during days 11.5–12.5 of pregnancy, at the time of neural tube closure, reduced the number of neurons in the motor nuclei of cranial nerves within the brainstem, without any other morphological changes in the brain. This was similar to the postmortem changes observed in the brain of an autistic child [155]. Later, the same group also found a reduction in the size of the cerebellar hemispheres and in the number of cerebellar Purkinje cells following an injection of 600 mg/kg VPA on day 12.5 of gestation in rats [156]. These findings were consistent with those observed in ASD in humans, such as a reduction in Purkinje cell numbers in the cerebellum and anomalies of the inferior olive in the brainstem and the deep cerebellar nuclei [157,158] (Table 5).

**Table 5 ijms-25-00390-t005:** VPA and autistic-like behavior in rodents.

Authors, Ref Number	Animals, Dose of VPA and Day of Treatment	Behavioral Outcome
**Schneider,** [148]	**Pregnant Wistar rats;** **single intraperitoneal injection of 600 mg/kg VPA on 12.5 day of gestation.**	ASD-like behavior appeared in VPA-exposed offspring at adolescence: lower sensitivity to pain, locomotor and stereotypic-like hyperactivity and exploratory activity and decreased social behaviors.
Win-Shwe, [149]	Pregnant Sprague-Dawley rats; intraperitoneal injection of 600 mg/kg of VPA on day 12.5 of gestation.	ASD-like behavior appeared in both adult male and female offspring: impaired sociability and impaired preference for social novelty.
Kawada et al. [150]	Pregnant ICR mice; intraperitoneal injection of 500 mg/kg VPA on day 9.5 of gestation.	ASD-like behavior: increased anxiety, reduced exploratory behavior and decreased social communication ability
Dos Santos et al. [151]	Pregnant BALB/C mice; intraperitoneal injection (IP) of 600 mg/kg VPA on day 12.5 of gestation.	ASD-like behavior in 5-month-old male and female offspring. Abnormal behavior in exploring novel environments and assessing risk. Higher impairment in females.
Chaliha et al. [144]	A systematic review of the literature exploring VPA’s effect on the presentation of ASD-like behavioral changes in rodents.	Main core ASD-like behaviors were defined: Social Impairment Repetitive Behaviors Cognitive Rigidity
Fereshetyan et al. [147]	Pregnant Sprague–Dawley rats; prenatally exposed to single IP injection with 500 mg/kg VPA at embryonic 12.5 day. Pups received 200 mg/kg of VPA on postnatal days 5–12. A battery of different behavioral tests was performed on PND30–40 (adolescence) and PND60–70 (adulthood).	ASD-like behavior detected in either prenatally or postnatally VPA-treated offspring: passive playing behavior, low preference for social interaction, increased repetitive behavior, locomotor hyperactivity, low exploratory activity and decreased anxiety. Behavioral changes were greater in adolescent rats than at adulthood.
Ornoy et al. [152]	Review: sex-related changes in the clinical presentation of ASD in children and in animal models of ASD-like behavior.	Behavioral, electrophysiologic and molecular sex-related differences found in genetic and non-genetic experimental animal models of ASD-like behavior.
Ornoy et al. [153]	Prenatal and early postnatal administration of a single dose of VPA 300 mg/Kg injected IP to ICR mice.	Prenatal and postnatal VPA-exposed offspring of both sexes had ASD-like behavior: more repetitive movements, higher anxiety and reduced memory in females. Reduced sociability, decreased working memory and normal locomotion in males.
Kotajima-Murakami et al. [159]	Subcutaneous injections of 600 mg/kg VPA to pregnant mice on gestational day 12.5.	ASD-like behavior: decreased social interaction.
Rodier et al. [155]	Pregnant rats; single IP injection of 350 mg/kg VPA on days 11.5–12.5 of pregnancy.	Reduced number of neurons in motor nuclei of cranial nerves within the brainstem.
Ingram et al. [156]	Pregnant rats; IP injected with 600 mg/kg VPA on day 12.5 of gestation.	Reduction in the size of the cerebellar hemispheres; reduction in the number of Purkinje cells.
Zhao et al. [160]	Cynomolgus monkeys (Macaca fascicularis), treated prenatally with 200 or 300 mg/kg VPA on gestational days 26 and 29.	ASD-like behavior: impaired social interaction, repetitive movements and more attention on nonsocial stimuli by eye tracking analysis.

ASD-like behavior was observed in our laboratory in mice injected with a single dose of 600 mg/Kg VPA on postnatal day 4. In addition to these behavioral studies, we also observed on day 60 changes in the expression of many genes, several of them already found to be associated with ASD in the SFARI list of genes [118,161]. This implies a possible epigenetic mechanism underlying the effects of VPA, probably related to the well-known inhibitory effects of VPA on histone deacetylase. There was also increased oxidative stress in the prefrontal cortex of VPA-treated pups. All effects including the changes in gene expression and behavior were sex-related [152].

Similar findings were observed following the injection of VPA on day 12 of gestation. Additionally, these mouse models showed evidence of abnormal but typical autistic-like calls on ultrasonic vocalization, abnormal EEG recordings and an imbalance between excitatory and inhibitory neurotransmission with repetitive and abnormal social behaviors typical to ASD patients reported and reviewed in [27,118,130,152,162,163]. The suggested mechanisms of action in the brain include increased glutamatergic neural density which leads to an excitatory/inhibitory imbalance, altered monoamines brain turnover, increased reactive oxygen species and epigenetic modifications [118,164].

The association of prenatal VPA exposure with autistic-like behavior was also demonstrated in non-human primates. Zhao et al. [160] carried out a study on cynomolgus monkeys (Macaca fascicularis) treated prenatally with 200 or 300 mg/kg VPA on gestational days 26 and 29. VPA-exposed juvenile offspring demonstrated impaired social interaction, repetitive movements and more attention on nonsocial stimuli by eye tracking analysis. Experimental animal models generally recapitulated the neurological and neurobehavioral effects of VPA in man, although animals seem to be more resistant to VPA. The neurobehavioral changes, including autistic-like behavior bear many similarities to those seen in humans, including sex-related effects.

## 13. VPA-Induced Changes in Gene Expression in the Brain of Animals

Studies utilizing animal models have shown that prenatal exposure to VPA can lead to changes in the expression of genes related to brain development, synaptic function and neurotransmitter signaling pathways, which are associated with ASD risk genes [118,161,165,166,167,168,169].

Feleke et al. [166] investigated how prenatal exposure to VPA influenced gene expression in term, day 21 fetuses. They studied fetuses from epileptic (Genetic Absence Epilepsy Rats from Strasbourg) and non-epileptic control rats to understand the impact of the genetic background. The analyses of genome-wide gene expression data showed that the majority of genes differentially expressed by VPA in epileptic pups exhibited similar changes to those observed in non-epileptic pups [166]. The investigators concluded that the altered pattern of gene expression triggered by VPA is independent of the genetic epilepsy background. Pathway enrichment analysis revealed that genes downregulated by VPA exposure exhibited enrichment for functional processes associated with the modulation of synaptic function and neuronal processes such as genes related to the glutamate receptor complex and neurotransmitter receptor activity, including genes for the regulation of insulin secretion [166].

Guerra et al. [169] found in C57Bl/6 male mice prenatally exposed to VPA that genes related to ASD were differentially expressed in the cerebellum during days PND1, PND10 and PND30. The genes comprised six different clusters that demonstrated dynamic changes during the different stages of postnatal cerebellar development. Some upregulated genes controlled synaptic functions and some downregulated genes controlled neurogenesis [169]. This finding provides evidence that progressive or regressive patterns of gene expression may reflect the processes occurring at the different stages of postnatal cerebellar development [170,171,172] such as the migration of neurons to the cerebellar cortex. Changes in gene expression during cerebellar development were associated with ASD-related genes and demonstrated overlap with 159 developmental-regulated genes annotated in either the SFARI database or/and Autism Gene Database (AutDB) [169].

Lenart et al. [165] analyzed 99 genes involved in excitatory glutamatergic and inhibitory GABAergic pathways in the cerebral cortex, hippocampus and cerebellum of Wistar rats that were prenatally exposed to VPA. The major differently expressed genes of glutamatergic pathways were observed in the frontal cerebral cortex. Twelve genes were upregulated, among them genes encoding the presynaptic glutamatergic proteins vGluT1 and mGluR7 and PKA that were elevated more than 100-fold. The number of differently expressed genes in GABAergic pathways in the cerebral cortex, cerebellum and hippocampus was relatively small [165] (Table 6).

Huang et al. [168] investigated the gene expressing analysis using a microarray in 50-day-old male rat hippocampi exposed to 600 mg/kg VPA on day 12.5. The majority of the 721 genes that were differently expressed were downregulated. The enrichment analysis of differentially-expressed genes in VPA-exposed rat hippocampi revealed associations with the plasma membrane, G-protein signaling, amine binding, and calcium signaling.

Zhang et al. [167] conducted whole transcriptome sequencing of the prefrontal cortex in adolescent male rats exposed to 600 mg/kg VPA on gestation day 12.5. They observed 3228 genes showing different expression levels and 637 genes exhibiting alternative splicing compared to the control group. These genes included those associated with neurological conditions like Huntington’s, Alzheimer’s and Parkinson’s diseases, as well as pathways linked to neurogenesis.

Kotajima-Murakami et al. [159] investigated whole mouse genome using microarray analysis of brains in adolescent male mice prenatally exposed on day 12.5 to 600 mg/kg VPA. Among differentially expressed genes in the brain, 2761 genes were upregulated, and 2883 were downregulated in comparison to saline-treated animals. In addition, an aberrant expression of genes induced by VPA was associated with the mTOR (mammalian target of rapamycin) signaling pathway. The VPA-treated offspring also demonstrated impaired social interaction behavior as tested with a three chamber test [159] (Table 6).

In our studies described above [118,161,173], VPA also induced sex-related significant changes in the expression of many genes in the prefrontal cortex, defined by Nanostring nCounter analysis of 770 neuropathology and neurophysiology genes. ICR mice were injected with a single dose of 300 mg/Kg VPA on postnatal day 4 and a gene expression analysis of prefrontal cortex was performed at 60 days. VPA-exposed offspring exhibited changes in the gene expression in a sex-dependent manner with more genes being changed in females. These changes were in pathways related to Huntington’s disease, Alzheimer’s disease, prostate cancer, focal adhesion, calcium and PIK3-signaling [161]. Enrichment in genes that are related to the Huntington’s disease pathway was associated with the mechanisms involved with the epigenetic modulation of DNA and HDAC functions. However, in the forebrain of day 1 newborn pups, prenatally exposed to 300 mg/kg of VPA on day 12 of gestation, we did not find significant changes in gene expression [173]. It is important to mention that these changes were inhibited by concomitant administration of SAMe. In addition, the treated mice typically demonstrated autistic-like behavior, with differences between sexes [152].

## 14. The Suggested Mechanism of the Teratogenic Action of VPA

### 14.1. Folate One-Carbon Metabolism (OCM) and Folate Deficiency

Theories on the mechanism of action of AED-induced malformations with anti-folate activity focused on the reduction in folic acid levels that often occurs after treatment with VPA and other AEDs. Indeed, VPA acts specifically as an antimetabolite to folic acid [44,146,174]. Reduced embryonic folic acid may disrupt gene expression, increase embryonic oxidative stress and induce changes in protein synthesis [127,174]. As folic acid antagonists generally produce birth defects [38], it would be consistent for anticonvulsant-mediated folic acid deficiency to result in fetal anomalies of the type observed in the offspring of VPA-treated women, i.e., NTD, cardiac anomalies and neurodevelopmental delay [48].

The current mechanism of VPA-induced congenital malformations is not precisely known. However, based on scientific data, several biological pathways by which VPA causes congenital malformations and neurodevelopmental abnormalities have been proposed. One of these pathways is one-carbon metabolism (OCM) where VPA depletes endogenous folate levels and inhibits the absorption of intestinal folate and transport of exogenous monoglutamic folates [142,175,176]

Through this pathway, VPA causes alterations in the glycine cleavage system, and blocks the dihydrofolate reductase and other enzymes involved in folate and SAMe cycles [142,177,178,179]. These effects of VPA may prevent the transfer of a one-carbon unit for the relevant cycles to progress, leading to abnormal methylation, phospholipid, polyamines, protein and nucleic acid synthesis [180,181]. It is reported that VPA, via the OCM, may also increase the oxidative stress burden, which will further induce major and minor congenital malformations [178,182,183,184,185]. Pregnant Wistar rats treated with different doses of VPA in mid and late gestational periods revealed alterations in several genes and enzymes of the OCM and folate transporter (folR1) [186]. Reynold and co-workers [175,187] demonstrated that VPA alters the expression of methylenetetrahydrofolate reductase and methyltetrahydrofolate reductase, and formyl hydrofolate reductase and dihydrofolate reductase. Alterations of the former are mediated in homocysteinuria while alterations of the later are involved in anti-tumorigenesis. VPA’s effect on formyl hydrofolate reductase inhibits the purine pathway responsible for protein synthesis [187]. The question of whether folic acid supplementation in pregnancy is effective in reducing the VPA teratogenicity is in debate.

The fact that folic acid administration, even in large doses, does not seem to prevent VPA-induced MCM may suggest that the above proposed mechanism of VPA teratogenic effects is of minor importance. Indeed, recent findings have continued to demonstrate the possibility that VPA teratogenicity is not primarily caused by its interference with folic acid metabolism, but by the alteration of several other pathways and mechanisms that play different significant roles in the OCM [6,27,187,188,189,190,191,192,193]. These current observations are evoking attention towards other components, including the important rate-limiting factors of OCM via which folic acid exerts its influence against the pathophysiology of VPA. (Figure 2).

### 14.2. Alterations in the SAMe Cycle and VPA-Induced Malformations 

S-Adenosine Methionine (SAMe), an FDA-approved food additive sold over the counter, has been found to have antagonistic actions against VPA. Being the primary and major biological methyl donor synthetized by all living organisms, it donates its methyl group to nucleic acid, proteins, and many other molecules for normal cell functioning [27]. SAMe serves as a precursor for the synthesis of polyamines, lipids, nucleotides, proteins, monoamine neurotransmitters, and glutathione (GSH) via three main interconnected metabolic pathways: polyamine synthesis, transmethylation, and transsulfuration [27]. The reaction between methionine and ATP catalyzed by methyltransferase (MTases) produces SAMe which is further catalyzed by other MTases, as a result of which *S*-adenosylhomocysteine (SAH) is produced. SAH is a competitive inhibitor of MTases and is essentially hydrolyzed to homocysteine and adenosine by *adenosylhomocysteinase* (AHCY). The ratio of SAM:SAH is used in determining cellular methylation potential whereby imbalance in this ratio will result in failures in methionine, SAMe, and reduced glutathione (GSH) synthesis, leading to homocysteinuria [179,194]. This represents the hallmark of aberrant methylation origin where VPA is a major culprit [195,196,197]. Women with NTD-affected pregnancies possess higher plasma concentrations of SAH with lower concentrations of plasma SAMe. Such metabolic arrangement corresponds significantly with reduced methylation capacity in consistently affected pregnancies [198]. Semmler and co-workers [178] showed that VPA altered the SAM:SAH ratio apparently leading to reduced hippocampal cell numbers, reduced brainstem volume, and impaired memory and learning. Ubeda et.al., [199] treated female Wistar rats with 400 mg/kg VPA and found significant reductions in methionine adenosyltransferase (MAT) activity at 1, 3, 6 and 9 h after VPA administration. They therefore proposed that VPA inhibits MAT activity and hampers SAMe synthesis, inducing alterations of methionine metabolism, and concluded that this could be the main mechanism underlying the pathophysiology of VPA teratogenicity [199] (Figure 2).

In support of this mechanism is the fact that SAMe administered concomitantly with VPA alleviated, in mice, the ASD-like behavior induced by VPA administration on postnatal day 5 [118]. In addition, SAMe corrected the changes in gene expression in the prefrontal cortex induced by this treatment [161].

### 14.3. The Inhibition of Histone Deacetylases (HDAC) 

There is growing interest and several clinical trials using VPA (and other HDAC inhibitors), especially in the field of cancer research, on the premise that HDACs are promising targets for therapeutic interventions intended to reverse aberrant histone acetylation states. Acetylation is tightly governed by the opposing actions of two large families of enzymes: histone acetyltransferases (HATs) and histone deacetylases (HDACs) [6,162,200]. The proper regulation of these two opposing enzymatic activities is of paramount importance for the normal control of development, the failure of which will result in disease conditions.

Another mechanism of VPA teratogenicity is this observed inhibition of HDACs leading to changes in gene expression induced in the neonate when VPA was administered on postnatal day 4, and in the fetus when given on prenatal day 12 of pregnancy [117]. Paradoxically, the epigenetic involvement of histone acetylation and deacetylation in numerous pathophysiological pathways implies that HDAC inhibition is capable of leading to the adverse effect of an aberrant global regulation of gene expression [200] (Figure 2).

The first report of VPA as an HDAC inhibitor was apparently by Menegola et al. [117] who, in HeLa cell lines, demonstrated VPA’s inhibitory activities on HDAC, its capability of halting the cell cycle to induce growth arrest and apoptosis. This inhibitory effect of VPA on proliferating cells presumably explains its teratogenicity affecting neural tube closure [117]. Histological analysis and RT-PCR assays to examine the expression of myocardial cell-related genes in mice exposed to VPA revealed that the transcriptional levels of heart development-related genes (CHF1, Tbx5 and MEF2), are significantly increased in the hearts of VPA-exposed mouse fetuses [136]. Hsie et al. treated chicken embryos and showed that valproic acid substantially downregulated genes folr1, IGF2R, RGS2, COL6A3, EDNRB, KLF6, and pax-3, resulting in NTD and other forms of congenital malformations [201,202]. Active hypomethylation is concurrently preceded by the active acetylation of H3 histones due to the direct inhibitory action of VPA on HDAC, thus opening up DNAs for easy access by the demethylases. The abnormal demethylation of DNA is apparently the trigger of changes in gene expression leading to various congenital anomalies (e.g., NTD) [203]. Pregnant rats, that were orally administered a single dose of VPA 400 mg/kg on GD 12/19 or repeated administration for 4 days (GD 9 to 12/GD 16 to 19), exhibited reduced placental expression of Folr1 and Mtr and increased expression of Dhfr corresponding with altered one-carbon metabolism and NTD, among other defects [186]. Several studies have reported important findings on the deleterious effect of VPA inhibition on HDACs in humans and animals, potentially causing major congenital defects [28,104,107,109,110,204].

Interestingly, the RNA sequencing analysis of human cortical neurons treated with VPA revealed that the genes showing differential expression were notably associated with functions such as mRNA splicing, mRNA processing, histone modification, and metabolism-related gene sets [205]. The analysis of differential transcript usage (DTU) revealed that VPA exposure brings about significant changes in DTU isoforms within the genes that are crucial for neurodevelopment and coincide with identified ASD-risk genes [205]. Altogether, these findings suggest that VPA inhibition on HDACs has a wider effect on chromatin remodeling, gene, expression and transcription.

It seems that the precise principal mechanism of VPA-induced teratogenicity is not yet established. However, several possible mechanisms were proposed, each one being partially proven by experimental data. As VPA is also an epigenetic modifier [162,173], to a certain degree antagonizing the effects of SAMe, it seems that the histone deacetylase inhibitory effects of VPA play a major role. A second plausible important mechanism is VPA’s folic acid-antagonizing effects. These mechanisms are especially important when considering the gene expression changes induced by VPA during a developmental prenatal and early postnatal period when immense changes in the epigenetic memory occur [152] (Figure 2).

### 14.4. Increased Oxidative Stress

Several AEDs, similar to diabetes and lead (Pb) teratogenicity cause a heightened oxidative stress burden on the developing fetus [4,206,207,208,209] which is implied as one of VPA’s mechanisms of teratogenic action [6]. Generally, the developing embryo lacks a fully functional antioxidant defense system in the early gestational age, and a consequent irreversible embryonic and fetal damage may result from the abnormal elevation of reactive oxygen-nitrogen species. The brain particularly lags behind other fetal organs considering the development of the body’s antioxidant system, and therefore is found to be more adversely affected in instances of oxidative stress [210]. VPA, when injected intraperitoneally on GD 8 to pregnant ICR mice induced marked fetal malformations and evidence of the formation of peroxynitrite and S-nitrosylation in addition to an altered GSH level that induced the apoptosis of neural tube cells and macrophages [111].

Al-Amin et al. [211] injected Swiss albino mice with a single I.P. injection of 600 mg/kg valproate on GD 12.5 and observed significant alterations in antioxidants, including increased levels of malondialdehyde (MDA), nitric oxide (NO), advanced protein oxidation product (APOP) and decreased glutathione (GSH), catalase (CAT) and sodium dismutase (SOD) activity. Increased brain oxidative stress induced by prenatal or early postnatal VPA administration was reported by several investigators using the VPA-induced model of autistic-like behavior [118]. Other investigators have reported VPA as pro-oxidative in humans, animals and cultured cells [28,139,182,183,184,202].

### 14.5. Mitochondrial Dysfunction

The dysregulation of mitochondrial metabolism has been implicated as a possible pathophysiology of abnormalities caused by VPA treatment [27,139]. Salimi reported a collapse of mitochondrial membrane potential that would undermine synaptic plasticity [108]. Investigations on the effects of VPA in undermining the functional integrity of mitochondria revealed various outcomes, including the conformational alteration of the inner membrane and transport chain proteins, ATP depletion and increased reactive oxygen species (ROS) [212,213]. A dose-dependent VPA-induced cristae distortion in pyramidal neurons that altered mitochondrial ultrastructure was suggested as a sign of an impairment of synaptic plasticity and neurogenesis in treated rodents [214,215]. It was also observed in pigs that VPA inhibits mitochondrial respiratory chain complexes I, leading to the asymmetrical distribution and fragmentation of cristae, raised oxidative stress and the enlargement of the mitochondrial matrix [216,217,218].

### 14.6. Inositol Depletion

Inositol plays a crucial role in major signaling pathways that influence diverse cellular functions and offers an interface between membranes and the cytosol, the coordination of endocytosis and vesicle trafficking. The depletion of inositol reportedly causes cranial NTDs in mouse embryos, in a scenario that folic acid deficiency did not- and this was explained by the fact that inositol is a key factor in cell shaping and rearrangement. Therefore, depletion in the levels of inositol may have severe adverse developmental consequences [219,220,221,222]. Similar to lithium, VPA has been reported to cause inositol depletion in yeast while in mice, both inositol and inositol-1-phosphate levels were decreased upon exposure to VPA, corresponding with a myo-inositol-1-phosphate (MIP) synthase activity, a vital enzyme in the process of glucose-6-phosphate conversion to inositol-1-phosphate which is apparently the rate-limiting step in de novo inositol synthesis [223]. Nevertheless, the dampening effect of VPA on inositol is advantageously utilized within the tenet of bipolar and mood disorder therapy [224].

## 15. The Prevention of VPA Teratogenicity in Animals in Relation to Congenital Malformations

### 15.1. Attempts to Minimize VPA Teratogenicity

Generally, if a substance is a suspected or proven teratogen, the advice is, whenever possible, to refrain from exposure. Often, this exposure cannot be prevented as the pregnant woman needs to continue her exposure to the teratogen for her or her conceptus’ health. A vivid example is the need to continue the treatment with anti-seizure medications in spite of their possible teratogenicity. Hence, the possibility to prevent the undesirable effects of a teratogen is of utmost importance [225]. A vivid example for such an approach is the prevention of NTD and other malformations by folic acid that has the backing of many animal and human studies.

Several investigators have made attempts to improve the efficacy and usage of VPA, by introducing different changes to the chemical structure of VPA in order to minimize the teratogenic effects while simultaneously retaining its therapeutic activity and potency [114,122,226,227,228,229,230]. So far, most attempts were not successful. However, with several VPA analogs now in clinical trial, there are possibilities that a potent and less teratogenic VPA congener will soon emerge.

Among the additional efforts to minimize the teratogenic effects are folate supplementation [127,190,231]. Other nutrient co-factors such as vitamins B6, B12, C, and E, choline, betaine, selenium, methionine and inositol are used in efforts to modify VPA teratogenicity [142,220], especially in cases that are not preventable by folic acid [108,143,189,220,223]. Pregnant rats treated orally with 500 mg/kg VPA and with high doses of vitamin C, selenium and grape seed extract showed that maternal death, rate of resorption and delayed fetal cervical ossification were significantly reduced in combined treatment groups compared to the VPA alone treatment [232]. Mice, injected intra-peritoneally with 70 mg/kg of L-methionine 30 min prior to the administration of 350 mg/kg VPA showed that the methionine treatment reduced the incidence of spina bifida [142].

Tiboni and Ponzano [105] treated pregnant ICR mice with 600 mg/kg of VPA and studied the rate of fetal weight, NTD (exencephaly) and skeletal malformations. Pretreatment, one hour in advance, with sildenafil at doses of 2.5–10 mg/Kg reduced the rate of exencephaly in a dose-related manner, with the highest dose reducing the rate of exencephaly induced by VPA from 28% to 10%. 

### 15.2. VPA and Folic Acid (FA) Administration in Animals

As mentioned above, the prevention of VPA-induced malformations in human by folic acid administration has relatively little effect. Controversial findings were also reported in animals. Several investigators reported that folinic acid, regardless of the route of administration, failed to reduce the incidence of VPA-induced exencephaly in rodents. Neither did it alter the teratogenicity of VPA on the brain, liver, or kidney and other embryonic tissues, suggesting that folate probably has no significance in the pathophysiology of VPA-induced congenital malformations [177].

Furugen et al. [186] reported that VPA altered mRNA levels of major carriers for folate, glucose, choline, SAMe and some hormones. They concluded that VPA targets the folate receptor FOLR1, and possibly other folate receptors, leading to the direct inhibition of placental folate uptake. Earlier, Fathe et al. [233], utilizing cell culture modeling, found that VPA is a noncompetitive inhibitor of high affinity folate receptors such as folate receptorα (FRα). Hence, VPA may interfere with folate metabolism by inhibiting glutamate formyl transferase, an enzyme that produces 5-formyltetrahydrofolate (folinic acid), thereby inhibiting methionine, choline and SAMe syntheses, leading to adverse epigenetic changes [175,187].

Several studies demonstrated the beneficial effects of folic acid in the alleviation of VPA-induced teratogenesis. For example, pregnant female albino rats were treated with varying doses of VPA and the offspring were sacrificed at PND 12. Electron microscopic examination of the brain showed the appearance of shrinkage, holes and absence of organelles, neurofilaments and myelin sheath. However, concomitant supplementation with folic acid retained the normal architecture and thickness of the Purkinje cells and cerebral cortex layers [234].

A single subcutaneous injection of 400 mg/kg VPA on GD 9 in mice induced NTDs, a twofold increase in p53, fourfold decreases in NF-κB, Pim-1, and c-Myb protein levels evaluated on GD 10. Additionally, VPA significantly increased the ratio of embryonic Bax/Bcl-2 protein levels. These observations were normalized by 3 days intraperitoneal treatment with 4 mg of FA [235].

Akimova et al. [195] administered a single dose of VPA to pregnant SWV mice fortified at pre-conception with FA and observed NTDs in all embryos from VPA-treated dams. FA supplementation could not prevent NTDs, rather it amplified metabolic profile differences that perturb lipid metabolism, and in combination with VPA-mediated carnitine depletion, evidently altered energy breakdown. Whole-mount immunostaining using an anti-neurofilament antibody performed on GD 10 embryos of pregnant ICR mice subcutaneously injected with a single dose of 400 mg/kg VPA revealed spinal nerve defects that included the disruption of the segmental pattern of dorsal root ganglia, among others that were not prevented by co-treatment with 3 mg/kg of FA [102].

Padmanabhan et al. [236] assessed the effective dosing regimen of folic acid for the prevention of VPA-induced exencephaly. They injected a single dose of 400 mg/kg of VPA into mice on gestation day (GD) 7 or 8, followed by co-treatments with a single dose of 12 mg/kg of FA or three doses of 4 mg/kg FA each. Multiple dosing of FA was associated with a sustained elevation of plasma level and caused an enhanced closure of the neural tube. This finding suggested that plasma FA level requires sustained elevation throughout organogenesis for enhanced protection against VPA-induced NTD [236].

Turgut et al. [237] used Fertile leghorn type chicken eggs as models for low and high doses of VPA and FA. The eggs were hatched for 24 h, injected with the designated compounds and then hatched until the 72nd hour. The high-dose VPA embryos had significantly higher cases of neural tube defects compared to low-dose, but FA did not significantly change this outcome [237].

Many studies in animals showed a beneficial effect of high doses of folic acid in the amelioration of VPA-induced teratogenicity in animals. However, considering several contradictory reports emanating from teratologic studies on the interaction of VPA and folic acid, it would be premature to mainly attribute the teratogenicity of VPA to folic acid depletion. There is still a need to systematically study other nutrients of the methyl cycle OCM like SAMe, choline and their associated enzymes.

## 16. The prevention of the VPA Induction of ASD in Preclinical Animal Models

Preventing or treating VPA-induced ASD-like behavior in animal models has been a subject of experimental research, primarily to better understand the mechanisms involved and to explore potential preventive measures. Some strategies and findings from experimental studies include supplementation with antioxidants [238,239], epigenetic modulator molecules [27,162], environmental enrichment [240], pharmacological [241] and genetic interventions [242].

The timing of pharmacological treatments in studies significantly impacts their effects and outcomes. Treating VPA-exposed offspring at different stages of development, whether it is prenatally, early postnatally, during adolescence or adulthood, can yield different results and treatment efficacy. Most reported pharmacological studies based on the strategy of giving treatments to VPA-exposed offspring at adolescence or adulthood reported that the beneficial effects of treatments were generally transient.

Nakhal et al. [238] evaluated the therapeutic effects of canagliflozin, the sodium-glucose cotransporter 2 (SGLT2) in the treatment of ASD-like behavior in adolescent rat offspring prenatally exposed to 500 mg/kg VPA on GD 12.5. Canagliflozin alleviated most ASD-like behavioral traits and decreased aggressive-like self-grooming behaviors in a dose-dependent manner. Moreover, canagliflozin treatment normalized the levels of glutathione, catalase and malondialdehyde, the oxidative stress protectors in the brains of the VPA-treated offspring.

Zhang et al. [239] reported that intraperitoneal injections of 150 mg/kg N-acetylcysteine (NAC, an antioxidant) given once daily for 4 weeks to VPA-exposed offspring at adolescence improved sociability and repetitive-like behaviors. In addition, it was demonstrated that NAC treatment inactivated the Notch-1/Hes-1 signaling pathway and recovered autophagy activity.

Yamaguchi et al. [240] demonstrated the beneficial effect of 4 weeks of enriched environment on ASD-like behaviors in 4-week-old CD1 mice prenatally exposed to VPA at embryonic day 12.5. The environmental enrichment improved anxiety-like behavior, social deficits, and cognitive impairment, but did not decrease locomotion, accompanied by enhancing dendritic spine function in the hippocampus.

Kumar and Sharma [241] investigated the effects of minocycline, a known modulator of retinoic acid signaling, in an ASD-like model induced by 500 mg/kg VPA given to pregnant rats on day 12.5.of gestation. Oral treatment with 25 and 50 mg/kg minocycline from postnatal days 21–50 normalized social interaction, spontaneous alteration, exploratory activity, intestinal motility, serotonin levels and prefrontal cortex mitochondrial complex activity.

Luo et al. [242] studied the role of the triggering receptor encoded TREM2 gene, which is expressed in the microglial cells and the resident macrophages in the CNS, in the induction of an ASD-like model via prenatal treatment with VPA. It was found that autistic-like behaviors in rat offspring prenatally treated with VPA were associated with downregulated levels of TREM2, microglial activation and altered TREM2 synapse integrity. The delivery of adeno-associated virus particles to the prefrontal cortex of VPA-exposed offspring induced TREM2 overexpression and subsequent improvement of autistic-like behavioral traits.

Some studies aimed to explore the potential of some vitamins such as vitamin A [243], vitamin D [244] or resveratrol [245] to prevent the neurodevelopmental abnormalities induced by VPA. In this context we should mention that the reduced concentrations of vitamin E or vitamin A in the blood were associated with ASD in human [246,247,248].

Liu et al. [243] injected pregnant rats with 600 mg/kg VPA on day E12.5 of gestation and administered Vitamin A (20,000 U/kg) by oral gavage from PND1 to PND7. Vitamin A supplementation reversed VPA-induced ASD-like behavior and increased the level of vitamin A in the serum of 5-day-old offspring compared to treatment with VPA only [243]. Vitamin A was administered during the period when a large percentage of the tissue-specific methylation patterns were generated in the brain [249]. Du et al. [244] found that early single treatment with vitamin D3 (80,000 IU/kg I.M.), in a 12-day-old rat offspring exposed to prenatal VPA, improved social interaction and repetitive behaviors.

Hirsch et al. [245] evaluated the effects of prenatal resveratrol, an antioxidant and anti-inflammatory molecule, on ASD-like behavior induced by VPA. Pregnant female Wistar rats were treated by a single intraperitoneal injection of 600 mg/kg VPA on embryonic day 12.5. Concomitantly, resveratrol was subcutaneously injected daily (3.6 mg/kg) from E6.5 to E18.5. Resveratrol treatment prevented the development of ASD-like behaviors and decreased the level of miR134–5p in rats induced by VPA.

In our studies [27,118,161], newborn ICR mice were treated on PND 4 with a single injection of 300 mg/kg of VPA, with normal saline (controls), or with SAMe that was given orally for 3 days at a dose of 30 mg/kg body weight starting from PND 5. SAMe administration to the VPA-treated mice reversed most ASD-like behavioral phenotypes [118], reduced the degree of oxidative stress in the prefrontal cortex [118], and normalized most of the VPA-induced gene expression changes [161]. Thus, as mentioned above, VPA and SAMe are both epigenetic modulators that seem to have antagonistic effects on the developing brain.

The different successful attempts to modify or ameliorate the ASD-like behavior in animal models were carried out only on the VPA-induced ASD-like model. They do not seem to reflect on human ASD. However, they highlight the fact that whenever you understand the etiology of a disease, it is easier to find a successful preventive measure or even cure.

## 17. Conclusions

Valproic acid seems to be the highest teratogenic drug among the AEDs and therefore, if possible, should be avoided during childbearing age, especially in pregnancy. Because NTD may be induced in the third week post fertilization, it may be too late to stop the medication when pregnancy is first diagnosed. Therefore, VPA-treated women at childbearing age should use contraceptives and stop the medication before any planned pregnancy. Moreover, due to its neurobehavioral negative effects even late in pregnancy (i.e., intellectual disability, ASD, ADHD and other neurodevelopmental disabilities), it seems advisable to change VPA to a different antiepileptic drug even during pregnancy if the alternative drug is effective. VPA significantly increases the teratogenic potential of other antiepileptic drugs. If there is no alternative to VPA treatment, the pregnancy should be considered high risk and must have proper follow-up, including appropriate antenatal diagnosis. It is also advised to use the lowest effective amount divided into three daily doses to minimize fluctuations of serum levels of VPA. Daily doses lower than 600 mg seem to have a low teratogenic effect. The addition of 4–5 mg/day of folic acid should also be considered, although its efficacy is somewhat in doubt. 

## Figures and Tables

**Figure 1 ijms-25-00390-f001:**
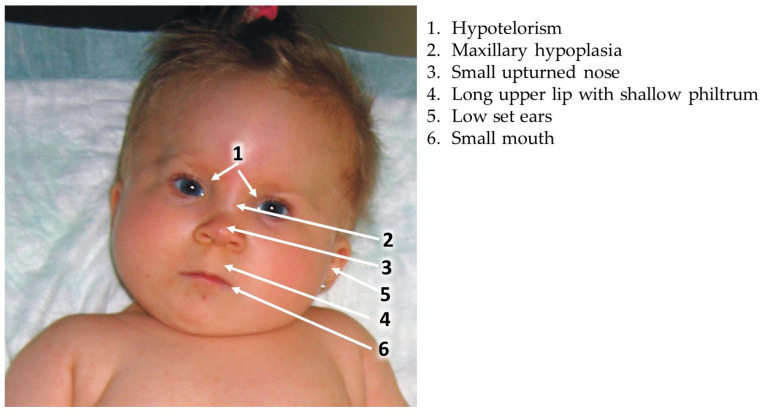
Typical facial changes in VPA syndrome.

**Figure 2 ijms-25-00390-f002:**
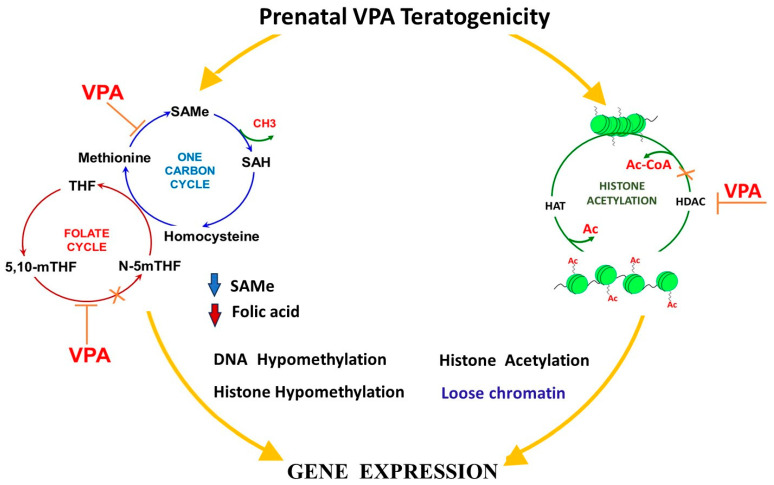
**Suggested mechanism of the teratogenic action of VPA-inducing epigenetic changes.** Several biological pathways by which VPA causes congenital malformations and neurodevelopmental abnormalities have been proposed. Among them, one-carbon metabolism (OCM) and inhibition of HDAC enzymes seem to be the main contributors to VPA teratogenicity. VPA blocks the dihydrofolate reductase and methyltransferases (MTases) enzymes that are involved in folate and SAMe cycles in one-carbon metabolism (OCM). Thus, VPA interferences in OCM resulted in depleted endogenous folate levels, SAM: SAH ratio imbalance and higher plasma concentrations of SAH with lower concentrations of plasma SAMe that were indeed observed in women with NTD-affected pregnancies. The decline in SAMe production, a critical methyl donor, contributes to a decrease in DNA methylation levels, subsequently increasing gene expression. VPA’s disruption of folate-dependent pathways and the subsequent impact on SAMe levels, in turn, influences DNA methylation, and gene expression, and potentially contributes to the observed developmental abnormalities and congenital malformations associated with VPA exposure during pregnancy. Similarly, inhibition of HDAC enzymes leads to enhanced histone acetylation and formation of loose chromatin, and therefore an increase in gene expression.

**Table 2 ijms-25-00390-t002:** VPA-induced neurocognitive deficits in children.

Authors, Ref. Number.	Main Findings	Comments
Bromley et al. [9]	Cochrane Library review demonstrating a high rate of neurodevelopmental problems following prenatal VPA.	A review that covers data from many studies.
Kini et al. [11]	A high percentage of the 63 VPA-exposed children had a high rate of dysmorphic features and a low verbal intelligence quotient.	VPA-exposed were compared to carbamazepine- and phenytoin-exposed children.
Bromley et al. [22]	Thirty-one individuals with fetal valproate syndrome; IQ was 19 points lower than controls and 26% had an IQ lower than 70.	The group consisted of children, adolescents, and adults.
Rithman et al. [24]	Comparison of neurodevelopment in 30 preschool children exposed to VPA and lamotrigine. Those exposed to VPA had lowest scores and increased rate of preschool ADHD.	Relatively small sample.
Cohen et al. [25]	Children prenatally exposed to VPA monotherapy had lower adaptive abilities and lower learning abilities, especially difficulties in auditory verbal and visual non- verbal functions, in a dose–response manner. They also had a high rate of ADHD.	Worse performance compared to carbamazepine, phenytoin and lamotrigine.
Christianson et al. [68]	Two twin pairs with fetal valproate syndrome and global developmental delay.	Case reports.
Moore et al. [63]	Children with fetal valproate syndrome of which 34 were exposed only to VPA and most had neurodevelopmental delay, many with some autistic features.	Among the first to describe autistic features resulting from prenatal VPA.
Dean et al. [65]	General notice that VPA exposure (and other AEDs induce learning difficulties and behavioral problems.	Gives criteria for the definition of AEDs syndrome.
Koch et al. [66]	Forty children exposed to AEDs. VPA-exposed had the worse neurodevelopmental outcome.	Relatively few exposed children.
Nicolai et al. [67]	A review defining the neurodevelopmental effects of prenatal exposure to AEDs. The worst of all is VPA.	A summary of several studies.
Daugaard et al. [76]	A total of 580 children exposed to VPA; hazard ratio of 4.48 for intellectual disabilities compared to controls.	Some increased risk for intellectual disabilities also in children exposed to carbamazepine, oxcarbazepine and clonazepam.
Dean et al. [69]	A total of 299 children exposed to AEDs, of which 47 were exposed to VPA. MCMs were found in 10.6% and developmental delay was found in 28%, the highest rate among all AEDs.	Many had typical dysmorphic features of the anticonvulsant drug syndrome.
Vinikainen et al. [70]	A need for educational support in 62% of 13 VPA-exposed children.	Small number of children.
Adab et al. [71]	Evaluated the educational risk of children prenatally exposed to AEDs. VPA exposure induced a high need of educational support, with an odds ratio of 3.4 compared to non-exposed.	Exposure to VPA had the worst effect in comparison to other AEDs.
Adab et al. [72]	Retrospective study of 41 school-age children exposed to VPA monotherapy. They had lower mean verbal IQ compared to those with exposure to other AEDs. A negative correlation was found between verbal IQ and dysmorphic features.	Retrospective study showing similar findings to prospective studies.
Tomson et al. [77]	Meta-analysis of prospective studies. Increased risk of cognitive impairment and autistic traits with VPA.	The risk for anomalies and intellectual impairment with VPA is the highest among several AEDs tested.
Shallcross et al. [78]	A comparison of neurodevelopmental outcomes in infants younger than two-years old with exposure to levetiracetam versus VPA. VPA induced in 40% of infants a DQ of less than 84.	Levetiracetam exposure did not reduce the DQ of infants.

**Table 4 ijms-25-00390-t004:** VPA-induced malformations in experimental animals.

Authors, Ref. Number	Dose of VPA and Mode of Treatment	Type of Malformations	Comments
Bold et al. [102]	Mice, 100 to 600 mg subcutaneously on GD 6 to 10.	32% NTD and 100% spinal nerve malformations	VPA caused dose- and age-dependent malformations when different doses were administered.
Di Renzo et al. [103,119]	Mice, 400 mg single I.P. VPA on GD 8.	50% abnormal somite formation, very high rate of axial malformations, 11.6% exencephaly	One study is focused on the somitogenesis of VPA-exposed embryos within 24 h of treatment and the other showed that methionine pretreatment before VPA worsens the abnormal outcome.
**Shafique & Winn**, [104]	ICR Mice, 400 mg subcutaneously on GD 9.	37.89% NTD with 18% average exencephaly	Indicated the role of epigenetics in the mechanism of VPA-induced teratogenesis.
Tiboni et al. [105]	CD-1 Mice, 600 mg intraperitoneal on GD 8.	28% exencephaly, 76% axial skeletal defect and 23% lethality	Reported the possibility of sildenafil citrate in decreasing skeletal malformations but not exencephaly.
Binkerd et al. [113]	Sprague-Dawley rats. 200–800 mg orally from GD 8 to 17.	skeletal malformations, ventricular septal malformations and great vessel and urogenital malformations	Human teratogen VPA was accessed in a dose-dependent manner by treating each mouse with a single dose of 100, 200, 400, 600 or 800 mg on different gestational days.
Lin et al. [114]	Swiss Vancouver mice, single intraperitoneal injection of VPA 1.8 or 2.7 mmol/kg on GD 8.	11.1% external malformations, 40% visceral malformations, cleft palate, heart and kidney malformations	Reported four genes (*Mtap2*, *Bmp8b*, *Stat3*, and *Heyl*) as candidate target genes of VPA-induced malformations.
Elmazar, M.M.A. and H. Nau [120]	NMRI mice, single 300 mg or 400 mg on GD 8.	12.9% exencephaly, 20% lethality with VPA alone	The cases of exencephaly rise to 42.5% when VPA is co-administered with trimethoprim.

**Table 6 ijms-25-00390-t006:** Changes in brain gene expression induced by prenatal or early postnatal VPA.

Authors, Ref Number	Description	Findings
Weinstein—Fudim et al. [161]	Nanostring nCounter analysis of 770 neuropathology and neurophysiology genes in the prefrontal cortex of offspring postnatally treated with 300 mg/Kg VPA on PND 4.	VPA induced changes in gene expression in a sex-dependent manner, with more genes changed in females. Gene enrichment in pathways related to Huntington’s disease, Alzheimer’s disease, prostate cancer, focal adhesion and calcium and PIK3-signaling.
Weinstein—Fudim, et al. [173]	Nanostring nCounter analysis of 770 neuropathology and neurophysiology genes in the one-day-old newborn pup forebrain, after prenatal exposure to 300 mg/kg of VPA on day 12 of gestation.	No significant changes in gene expression were found.
Lenart et al. [165]	A total of 99 genes related to excitatory glutamatergic and inhibitory GABAergic pathways in cerebral cortex, hippocampus and cerebellum from Wistar rats prenatally exposed to VPA.	Genes encoding the presynaptic glutamatergic proteins vGluT1 and mGluR7 and PKA were elevated more than 100-fold, mostly in the frontal cerebral cortex. Gene expression changes related to GABAergic pathways were relatively small in all three brain areas.
Kotajima-Murakami, et al. [159]	Whole mouse genome was investigated by microarray analysis in the brains of adolescent male mice prenatally exposed on day 12.5 to 600 mg/kg VPA.	Among differentially expressed genes in the brain, 2761 genes were upregulated, and 2883 were downregulated. Differentially expressed genes by VPA were associated with the mTOR (mammalian target of rapamycin) signaling pathway.
Feleke et al. [166]	Genome-wide gene expression analysis of 21-day fetuses brains from epileptic (Genetic Absence Epilepsy Rats from Strasbourg) and non-epileptic control rats.	VPA induced an altered pattern of gene expression independent of the genetic epilepsy background. Differentially expressed genes were related to synaptic function and neuronal processes (glutamate receptor complex and neurotransmitter receptor activity and regulation of insulin secretion).
Zhang et al. [167]	Whole transcriptome sequencing of the prefrontal cortex in adolescent male rats exposed to 600 mg/kg VPA on gestation day 12.5.	A total of 3228 genes were differentially expressed, with 637 genes exhibiting alternative splicing. The changes were associated with genes associated with Huntington’s, Alzheimer’s, and Parkinson’s diseases and associated with pathways related to neurogenesis.
Huang et al. [168]	Microarray analysis of 721 genes in 50-day-old male rats hippocampus exposed to 600 mg/kg VPA on gestational day 12.5.	Most differently expressed genes were downregulated. These genes were associated with the plasma membrane, G-protein signaling, amine binding and calcium signaling.
Guerra et al. [169]	Gene expression analysis in PND1, PND10 and PND30 cerebellum of C57Bl/6 male mice prenatally exposed to VPA.	Differentially expressed genes of six different clusters demonstrated progressive or regressive patterns of gene expression during the different stages of postnatal cerebellar development. Genes were related to synaptic functions or neurogenesis. They were associated with ASD-related genes overlapping with 159 developmentally regulated genes appearing in either the SFARI database or/and the Autism Gene Database (AutDB).

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
