# Peer review of "Valproic Acid in Pregnancy Revisited: Neurobehavioral, Biochemical and Molecular Changes Affecting the Embryo and Fetus in Humans and in Animals: A Narrative Review"

_ijms, 2023, doi:10.3390/ijms25010390_

Round 1
Reviewer 1 Report
Comments and Suggestions for Authors
The manuscript is well organized and presented.
Some minor points are suggested to highlight the originality of this review, as this topic has been revised by other authors.
a) Please edit personal comments. Those can be presented as crytical analysis, but some paragraphs seem difficult to follow as they are from a personal perspective.
For example: The contrasting faces of VPA – many therapeutic benefits as opposed to the high percent of rather severe abnormalities if taken during pregnancy remind us of the well - known Novella written by Robert Louis Stevenson and published in 1886: “Strange Case 60 of Dr. Jekyll and Mr. Hyde”. It should be clearly related with the contrasting situation in VPA administration.
Also, at the end of each section, the paragraphs starting "In summary".
b) Conclusions should be carefully edited. All revised data should be commented on this section, and a legend regarding the recommended posology should added (for taking it cautelously, as the antiseizure therapy should be personnel).
Comments on the Quality of English LanguageMinor details
Author Response
We thank the reviewer for the instructive comments.
We addressed all comments, and the changes are marked in red in the text.
- “please edit personal comments”
We edited the personal comments based on the cited studies, as suggested. For example, we added to the citation of “Stevension’s Novella” on page 2 lines 61-62 why VPA reminds us this Novella; “Where Dr. Jekyll is the good and perfect man (very effective drug) but if taken during pregnancy it may become Mr. Hyde, the ultimate evil and a murderer”
We omitted the words: In summary and started the summary differently.
- As suggested by the reviewer, we made several changes in the conclusions that can be observed in the text.
Reviewer 2 Report
Comments and Suggestions for Authors
In this review by Ornoy et al., the authors explore studies on the effects of the teratogenic valproic acid in pregnancy in humans and animals.
The article is very comprehensive and includes all relevant information on the topic from the 1980s to the present day.
There hasn't been a review on the subject in some time and unlike others, it contains novel information ex. valproic acid and ADHD.
Every sub-section is summarised at its end, which is very convenient and makes the long article easy to follow.
The tables are very neat and help to follow and revise the extensive content of the text.
An illustration of "Valproate syndrome" would be a helpful and interesting addition to the article in order to better highlight the typical craniofacial dysmorphism, skeletal and other malformations.
All in all, it is an extremely interesting and well-written review. I enjoyed reading it and I'm sure other readers will too.
Comments on the Quality of English Language
I have noticed minor errors. For example, in line 96 the authors used the wrong tense.
Author Response
We thank the reviewer for his positive and important comments.
As suggested by him, we added a figure (figure 1) that shows the typical craniofacial changes of VPA –induced facial dysmorphism.

Reviewer 3 Report
Comments and Suggestions for Authors
Ornoy et al. They present a narrative review of the literature on a high-stakes topic. The manuscript is capable of bringing together all the information necessary to have a specialized vision. The distribution of the epigraphs is correct and the tables presented are adequate. However, some changes are necessary:
-The title of the manuscript must include the word "Narrative Review"
-Writing should be improved with an expert in English language editing.
-The authors must include a more specific section on clinical trials in progress, as well as those that have been negative.
-Section 16 must be changed and expanded. This section should be titled "preclinical models." It must be extended and improved. Each model must be discussed and described.
.A figure on the mechanism and physiopathology during pregnancy should be included and integrated into the text of the manuscript.
-The paragraphs that are titled as summary must be integrated into the text of each epigraph.
-Authors must include a graphic summary.
Comments on the Quality of English Language
Moderate editing of English language required.
Author Response
We thank the reviewer for the important and instructive comments.
- We added to the title the words: “A narrative review”
- We proof read the manuscript to improve the English
- Add a specific section on clinical trials in progress:
We added at the beginning of section 3 the names and web addresses of the larger registries on AEDs in pregnancy, as follows: “There are a number of ongoing international registries regarding the use of AEDs during pregnancy. Among the largest are: The North American Antiepileptic Drug registry (AED Pregnancy Registry - https://www.aedpregnancyregistry.org); the North American Antiepileptic Drug Pregnancy Registry by the National Institutes of Health (.gov) (https://pubmed.ncbi.nlm.nih.gov ), the EURAP - International Registry of Antiepileptic Drugs (https://eurapinternational.org) , the [MI] Medicines and Pregnancy Registry – Antiepileptic use NHS Digital in the UK (https://digital.nhs.uk › publications › statistical › antie) and the Australian Pregnancy Register For Women on Antiepileptic Medications (ABN: 38 128 668 797 ). Each of these registries had published their findings on the teratogenicity of VPA and many of their studies are discussed by us together with studies by other investigators (Table 1).
To the best of our knowledge, all registries found high VPA teratogenicity with negative neurodevelopmental effects”.
- Section 16 must be changed….
We expanded section 16 as requested; named it “Prevention of VPA induction of ASD in preclinical animal models” as suggested and described some of the animal models in more detail.
- A figure on the mechanism of pathophysiology of VPA during pregnancy should be added
We added a figure (Figure 2) describing the VPA proposed mechanism of teratogenic action as related to one carbon metabolism, Folic acid and SAMe. This is added to section 13 – 1,2 and 3. We did not think that a graphical presentation of the other proposed mechanisms (i.e. oxidative stress, inositol depletion and mitochondrial dysfunction) should be added as these are simple proposed mechanisms.
- The paragraphs that titled the summary must be integrated…
This was done
- A graphic summary:
Done